# Remaining Useful Life Prognostics of Bearings Based on a Novel Spatial Graph-Temporal Convolution Network

**DOI:** 10.3390/s21124217

**Published:** 2021-06-19

**Authors:** Peihong Li, Xiaozhi Liu, Yinghua Yang

**Affiliations:** College of Information Science and Engineering, Northeastern University, Shenyang 110819, China; 1970640@stu.neu.edu.cn (P.L.); yhyang@mail.neu.edu.cn (Y.Y.)

**Keywords:** RUL, ASRMS, graph convolution, temporal convolution

## Abstract

As key equipment in modern industry, it is important to diagnose and predict the health status of bearings. Data-driven methods for remaining useful life (RUL) prognostics have achieved excellent performance in recent years compared to traditional methods based on physical models. In this paper, we propose a novel data-driven method for predicting the remaining useful life of bearings based on a deep graph convolutional neural network with spatiotemporal domain convolution. This network uses the average sliding root mean square (ASRMS) as the health factor to identify the healthy and degraded states, and then uses correlation coefficient analysis on the hybrid features of the degraded data to construct a spatial graph according to the strength of the correlation between the obtained features. In the time domain, we introduce historical data as the input to the temporal convolution. After the data are processed by the spatial map and the temporal dimension, we perform the prediction of the remaining useful life. The experimental results show the accuracy of the method.

## 1. Introduction

Prognostic and health management (PHM) is a critical technology to ensure the safety and reliability of equipment, which has achieved fruitful theoretical results in the past decades and has been widely applied [1,2,3,4,5]. RUL prediction has a guiding value for maintenance decisions and equipment ordering, and has long been the basis and core of PHM technology. Rolling bearings are key equipment of modern industry, the prediction and diagnosis of their condition has vital significance. This kind of equipment under the combined effect of internal and external factors, performance, and health status will inevitably show a trend of decline. When the degradation reaches a certain level, they are unable to perform its normal tasks and functions [6,7,8,9]. A prediction method is needed to determine the health status of rolling bearings during operation, which can be used to indicate impending failures and provide more time for maintenance of the equipment [10,11,12].

In recent years, RUL prediction methods have been characterized by diversity, hybrid, and complexity. Zhang [13] et al. classified the research methods for RUL prediction of rotating machinery into three main categories: failure mechanism-based methods [14], data-driven methods [15], and a combination of both [16]. The failure mechanism-based approach can predict the remaining life of the equipment more accurately, but it requires a lot of physical knowledge, and this approach is difficult to be applied in practice. On the other hand, the complexity of vibration signals makes it too costly to perform accurate failure mechanism modeling. In contrast, data-driven approaches can infer correlations and cause–effect relationships hidden in the data, as well as learning potential trends from the data. Consequently, data-driven modeling-based approaches have seen a large increase in their number of applications as a result of their accurate predictions. Methods that combine the two make full use of the advantages of both, but the process is more complex and thus these methods are rarely used [17].

In practical engineering, the degradation processes of different devices are different and unknown, and the inappropriate selection of degradation models will seriously affect prediction accuracy [18]. Machine learning (ML)-based methods can overcome the problem of unknown degradation models. According to the depth of ML models, ML-based data-driven life prediction methods are divided into two categories: shallow ML-based and deep ML-based methods. The shallow ML methods mainly include multi-layer perception (MLP), radial basis function (RBF), extreme learning machines (ELMs), etc. The method of MLP network trained by BP algorithm is usually called BP neural network. The literature [19] earlier used the backpropagation (BP) algorithm for RUL prediction. RBF neural network has a single hidden layer with three feedforward networks and can approximate nonlinear functions with arbitrary accuracy [20,21]. Chen et al. [22] proposed a multivariate gray-based RBF hybrid model for RUL of industrial equipment. Liu et al. [23] usually extracted multi-scale time-domain, time-frequency-domain features, and introduced a two-layer ICA algorithm to reduce dimension, then used ELM network to predict the downscaled data with satisfactory results.

It is worth noting that the present RUL prediction faces the problem of identifying the healthy and degraded states of the equipment, so the selection of a suitable starting degradation point plays a significant role in the prediction accuracy of the model. Pan et al. [24] proposed a two-step prediction method based on ELMs to identify the healthy and degraded states by constructing HI curves. Once the bearing is determined to be in the degradation stage, the prediction is implemented immediately to achieve high accuracy. Liang et al. [25] extracted hybrid features from the data and then used recurrent neural network (RNN) to construct health indicter (HI) curves for the extracted features, which achieved good prediction results. As a variant of RNN, Hou et al. [26] adopt the method of comparing the similarity between different devices and use BI-LSTM for the construction of HI factors to prediction.

Recently, deep neural network-based approaches have also achieved excellent performance. Deep learning, a new technique developed in recent years, provides a new approach for training large amounts of data with its powerful feature extraction capability. With the accumulation of neuron layers, deep learning networks are able to extract representative features of the original signal compared to shallow ML networks [27]. Chang et al. [28] proposed a deep belief network (DBN)-based modeling of the bearing degradation process using a particle swarm algorithm for optimal parameter search, showing a more powerful capability than the traditional RBF. Convolution neural networks (CNNs), as a class of classical feedforward neural networks [29], mainly consists of several convolutional and pooling layers, whose purpose mainly lies in building multiple filters to extract features hidden in the monitoring data level by level. For CNNs, the convolutional layer uses the original data input to convolve multiple local filters, and subsequently the pooling layer is able to extract the most important features at a fixed length, generally using the maximum pooling function [30]. Li et al. [31,32,33] proposed CNN-based remaining lifetime prediction and combined generative adversarial networks (GANs) [34] with deep CNNs for degenerate and healthy state distinction, which has been substantially improved in the prediction accuracy. Yao et al. [35] proposed a prediction model combining 1D-CNN and simple recurrent unit (SRU) networks, using 1D-CNN as a feature extractor, and then using SRU to predict the constructed data.

On top of that, graph-based convolutional neural networks have made remarkable achievements in natural language processing [36,37,38,39]. Compared with discrete CNNs, graph neural networks are able to extract deep-level features for high-dimensional data without destroying the topology of data. The simple vibration signal is low-latitude, but it possesses high-dimensional features. In the high-dimensional space, the data cannot be measured using Euclidean distance, and the convolutional neural network represented by CNNs use Euclidean distance to measure the relationship between the data, so it sometimes causes relatively large errors. In view of this, this paper proposes a new spatiotemporal composition method for bearing data, which introduces graph convolution with temporal convolution [40,41] for the prediction of RUL of bearings and proposes a method to confirm the first degradation point. The proposed model uses the authoritative bearing degradation dataset for performance testing.

In Section 2, this paper describes the preliminaries, including the confirmation of the first degradation point, graph convolution, temporal convolution, and the graph structure of bearings. The proposed method is demonstrated in Section 3, and the experimental validation is presented in Section 4. The whole paper is summarized in Section 5.

## 2. Preliminaries

### 2.1. Average Sliding Root Mean Square Value

The international standard ISO 20816-1-2016 [42] provides clear criteria for the degradation of vibrating equipment, and the degradation process of bearings is divided into four stages: stage A is the stage of healthy operation of newly deployed equipment; stage B is the stage in which vibrating equipment can operate for a long time; stage C is usually considered to be the stage in which equipment cannot operate for a long time; Stage D is considered to be the degradation stage, where the equipment is considered to be operating in a very dangerous condition. In view of this, root mean square (RMS) can be a good indicator of the degradation of the vibration signal [43] According to ISO 20816-2016 standard, for a small motor drive bearing, its operating state switches from state A to state B when the RMS value is larger than 0.71. Therefore, we choose when the RMS value is larger than 0.71 as the beginning of bearing degradation. The RMS curves of the bearings are shown in Figure 1.

We have selected four typical bearing degradation processes, and from the figure, we can see that not all bearings have a monotonic and relatively smooth degradation state. Some bearings may have RMS values exceeding 0.71 at the beginning of operation due to other reasons such as noise, and individual outlier points [44] may be perturbed in the middle of operation and greater than 0.71. It is obviously inappropriate to take such RMS values as the first degradation point (FDP), which will have a huge impact on the prediction of RUL. To address the effect of individual outliers on the first degradation point, we propose the average sliding root mean square (ASRMS) approach to determine the FDP
(1)$$\mathrm{ASRMS}=\frac{1}{N}\overset{N}{\underset{M=1}{\sum }}\sqrt{\frac{\sum_{i=1}^{M}x_{i}^{2}}{M}}$$
where $x_{i}$ denotes the original vibration value and N represents the number of sample points. The RMS values after the smoothing operation are shown in Figure 2. By observation, we find that the curve of ASRMS is smoother than that of RMS, and it can well avoid the interference of outlier points to the judgment of the FDP. For example, the RMS curves of bearings 2-4 and 3-1 have individual sampling points greater than 0.71 in the early and middle periods, and it is not correct to judge such outliers as the FDP, but after using the smoothing operation, the effect of outliers is eliminated. In addition, the ASRMS curve has good monotonicity, which can describe the degradation process of the bearing well. In summary, the ASRMS method proposed in this paper can well identify the health and degradation states of bearings.

### 2.2. Spatial Graph-Temporal Convolution

#### 2.2.1. Spatial Graph Convolution

The graph convolutional network (GCN) demonstrates good performance in solving data with graph structure. GCN is generally divided into spatial domain convolution and spectral domain convolution. The spatial domain convolution method generally operates directly on the central node and the adjacent nodes for feature extraction by certain rules [45]. Unlike spatial domain convolution, spectral convolution uses the eigenvalues and eigenvectors of the Laplacian matrix of the graph, and this method performs the convolution operation in the frequency domain using the Fourier transform of the graph. Based on the way the bearing data are constructed, we choose spatial convolution as our way of extracting features. The spatial convolution operation is shown in Equation (2)
(2)$$F_{i}=\overset{K}{\underset{k_{i=1}}{\sum }}\theta_{k_{i}}{L\;A}_{k_{i}}$$
where $F_{i}$ represents the ith output of the feature mapping; K indicates the size of the convolution kernel; $k_{i}$ imply the ith convolution kernel; $\theta_{k_{i}}$ is the weight parameter matrix, similar to the original convolution operation, given a weight vector for the input data; L means the input of the previous node or the original data, and $A_{k_{i}}$ is the diagonalized matrix of original adjacency matrix $\tilde{A}$. The primitive adjacency matrix $\tilde{A}$ is an O×O semi-positive definite matrix, where O are the number of nodes. Each element represents the presence or absence of a node connection, with nodes that do not have a connection having a value of 0 and nodes that have a connection having a value of 1.

#### 2.2.2. Temporal Convolution

The bearing signal collected by the sensor is a time sequential signal, we not only need to extract the features of the signal in the spatial dimension, but also in the temporal dimension. In view of this, we introduce temporal convolution to make the data correlated with each other in time dimension. After the spatial features of the bearings are extracted by the graph convolutional network, we use 2D convolution to extract the current temporal information. Equation (3) describes the temporal convolution formula for layer ,l at time  t
(3)$$O_{t}^{\left(k,l\right)}=\overset{K}{\underset{k=0}{\sum }}W_{k}a\left(f_{k}x_{t-\left(K-k\right)d}\right)$$
where $O_{t}^{\left(k,l\right)}$ represents the output of the lth layer and the kth convolutional block at moment t; a denotes the non-linear activation function; $W_{k}$ indicate the parameters of each convolution kernel $f_{k}$; $x_{t}$ serves as the input sequence after the graph convolution, k means the number of convolution kernels; d is the expansion rate of the convolution kernels; and (K−k)d shows the size of the perceptual field.

### 2.3. Spatial Domain Construction of Bearings

The complexity of the signal of vibration entails it difficult to fully characterize the degradation process of the bearing using a single time-domain or frequency-domain feature, which creates a great problem for the prediction of the bearing RUL. On the other hand, the feature extraction of traditional CNNs will destroy the intrinsic topology of complex data, while graph convolution will extract the features of the data also preserving the topology of high-dimensional data. Therefore, in this paper, we propose a novel construction of graphs for vibration signals to extract features to be able to predict RUL well.

In this paper, the original signal is extracted by time and frequency domain features, and max, min, peak to peak value, var, std, mean, rms, skew, kurtosis, mean-abs, 10 features are selected as input nodes, as shown in Figure 3. Figure 3 shows the tendency of different features, in order to analyze the strength of correlation between each feature, that is, to check the connection of each node in the graph, Pearson correlation coefficient analysis is used in this paper, as shown in Equation (4).
(4)$$\Upsilon \left(X,Y\right)=\frac{\mathrm{Cov}\left(X,Y\right)}{\sqrt{\mathrm{Var}\left[X\right]\mathrm{Var}\left[Y\right]}}$$
where the X,Y indicate extracted feature respectively.

We use the heat map to represent the correlation coefficients to obtain an O×O correlation matrix, where O means the number of extracted features. The two nodes with correlation coefficient greater than 0.75 are considered to be connected and the corresponding value in the adjacency matrix A is set to 1. The two nodes with coefficients less than 0.71 are unconnected and the corresponding value in A is set to 0. Thus, we obtain an O×O original adjacency matrix.

The input dimension of the final network is M×C×T, where M means the number of sample points, C is the number of nodes, and T denotes the time length of each sample point. Figure 4 shows the example of heat map of the correlation analysis, and the correlation matrix between the obtained features becomes the bearing graph structure shown in Figure 5 after the spatial feature mapping. The mean node has less correlation with other nodes, thus there are only nine nodes in Figure 5. The spatial convolution operation of the bearings is shown in Figure 6.

## 3. Proposed Method

### 3.1. Structure of the Proposed Spatial Graph-Temporal Convolution Network

Figure 7 shows the structure of spatial graph-temporal convolution network (SG-TCN). First, the training data and test data are processed by spatial composition and used as the input of the deep neural network. A spatial graph convolution layer and a temporal convolution layer are included in each layer, and the original data are convolved by the graph to effectively extract information related to other nodes at each node, which avoids the defects of using a single feature and failing to characterize the whole degradation process.

Secondly, the data after graph convolution is subjected to the operation of temporal convolution. The information contained in the time sequences can have a considerable impact on the prediction of RUL, so the feature extraction on the temporal dimension of the time sequences signals is performed using temporal convolution network (TCN).

After that, the data that have passed through SG-TCN use the Dropout technique [46], in order to keep the output from overfitting. Generally, the Dropout is set to 0.5. There are five SG-TCN convolution layers, and each layer of graph convolution uses a convolution kernel of convolution size 3 × 1, and the temporal convolution uses a convolutional kernel of size 5 × 1 with a convolution step of 1. After each SG-TCN operation, using the residual module to connect the input to the output to ensure that the necessary features are not lost.

The features extracted after the multilayer network then perform a maximum pooling operation [47], which is used to retain the most salient features. The maximally pooled data is passed through a flatten layer and a fully connected layer containing 256 neurons. The fully connected layer is connected to an output neuron, which executes RUL prediction.

The activation function used for each convolutional layer is rectified linear units (ReLU) [48], which is well suited to avoid the gradient disappearance problem. This study also uses the all-0 padding technique [49] to ensure that the data dimension does not change during the convolution operation.

### 3.2. Flow Chart of the Rul Prognostics Method

Figure 8 shows the flow chart of the proposed method. Recent studies have shown that bearings do not degrade from the beginning, so prediction of the entire life cycle is not necessary. Therefore, the ASRMS method is used to identify the degradation characteristics and health characteristics of the bearing’s life cycle. The prediction starts when the bearing operates to the first degradation point (FDP). 

The training sets and test sets are prepared next. This study focuses on the degree of degradation of the machinery and the remaining useful life corresponding to different degrees of degradation. The training labels were used to train the training data sets in the form of RUL percentages. The RUL was defined as the value of the remaining life left from the FDP point and the RUL percentage was defined as the percentage of the RUL value to the full RUL value at the current moment.

After that, the training data are trained with the established spatiotemporal graph convolutional depth network. The bearing time sequences are extracted using temporal convolution, and the output values are matched with RUL labels. When the training sets are completed, the test data can be put into the network for testing.

### 3.3. Training Details and Evaluation Indexes

The PyTorch deep learning framework is used in this experiment. The backpropagation algorithm [50] was used for training with stochastic gradient descent optimizer (SGD) and Nesterov momentum was set to 0.1. Batch size was set to 32, weight decay was set to 0.0001, and learning rate was set to 0.0001. Mean squared error (MSE) is used as the loss function for back propagation, as shown in Equation (5). A total of 100 epochs are trained in this experiment.
(5)$$\mathrm{MSE}=\frac{1}{N}\overset{N}{\underset{i=1}{\sum }}{\left(x_{i}-{\hat{x}}_{i}\right)}^{2}$$
where N is the total number of samples, $x_{i}$ represents the actual RUL value, and ${\hat{x}}_{i}$ means the RUL value predicted by the deep learning network. In this paper, the mean absolute error (MAE), root mean squared error (RMSE), and the remaining lifetime percentage error are used to evaluate the network performance as shown in Equations (6)–(8). The meaning of the parameters in Equations (6)–(8) are as same as in Equation (5)
(6)$$\mathrm{MAE}=\frac{1}{N}\overset{N}{\underset{i=1}{\sum }}|x_{i}-{\hat{x}}_{i}|$$
(7)$$\mathrm{RMSE}=\sqrt{\frac{1}{N}\overset{N}{\underset{i=1}{\sum }}{\left(x_{i}-{\hat{x}}_{i}\right)}^{2}}$$
(8)$${\%\mathrm{Er}}_{i}=100\times \frac{x_{i}-{\hat{x}}_{i}}{x_{i}}$$

## 4. Experiments

### 4.1. Data Sets Description

The experimental data provided by the PRONOSTIA [51] experimental platform was used to describe the degradation of the bearing throughout its operational life, as shown in Figure 9. The measurement part of the platform contains two types of sensors, vibration, and temperature. The vibration sensor consists of two accelerometers, which are placed on the horizontal and vertical axes, respectively. In this paper, the data generated by the horizontal axis accelerometer is used for the experiments. The sampling frequency of the accelerometers is 25.6 KHz, and the sampling is executed every 0.1 s. Each sample—i.e., each acc file—contains 2560 sampling points, and the sampling is conducted every 10 s. 

During the operation of the bearings, a pressure of 4KN–5KN was applied to the loading part of the RONOSTIA experimental platform in order to be able to accelerate the degradation process of the bearings. After degradation acceleration, the life cycle of the bearing varies from 1–7 h, so it is very competitive to predict its RUL. According to the pressure and speed of the load, the operating conditions of the bearings are divided into three types, as shown in Table 1. In order to avoid causing damage to the experimental platform, the experiment was stopped when the amplitude of the vibration signal exceeded 20 g. In this study, when one bearing is used as the test set, the rest of the bearings are used as the training set.

### 4.2. Experimental Results

This section focuses on the prediction of the RUL values of the tested bearings using the proposed method. The superiority of the proposed method is shown by comparing it with other methods. The RUL prediction curves for the test bearing are shown in Figure 10. Since in the prediction curve usually contains local fluctuations, which have a relatively large effect on the degradation estimation of RUL. Accordingly, we take a smoothing operation for each RUL curve, as shown by the bolded green line in Figure 10. In addition, the smoothed curves for bearings 1-2 are not bolded because the predicted fluctuations for bearings 1-2 are not as large as those for the other bearings, so there is no particular need to highlight the degradation trend.

By observing the RUL prediction curve, we can find that the prediction curve can fit the RUL label well with obvious monotonic decline despite the slight local fluctuations. In addition, the prediction of the later stage of bearing degradation state is more important than the prediction of the earlier stage. The RUL prediction curves in Figure 10 converge sharply when the degradation is close to failure to avoid delayed prediction. Accurate prediction at the end of degradation can ensure the reliability and safety of equipment operation.

We can observe from the figure that the RUL curves do not exactly fit the training labels, which is due to the fact that most of the bearings run non-linearly in the degradation phase. On the other hand, the training with linearly degraded labels does not match well with the input data, therefore, the curves do not always lean into the degraded labels. Table 2 shows the prediction percentage errors of the proposed method. The number in each row in Table 2 implies the value of actual RUL and predicted RUL for each moment. In Table 2, it is observed that each bearing has a relatively small prediction percentage error at the beginning of the prediction and a relatively large error at the end of the prediction. This is because the RUL values in the late prediction period are smaller, thus causing a larger percentage of error.

### 4.3. Comparison with Other Methods

In this paper, we use SG-TCN to extract deep features of bearings for prediction. To verify the effectiveness of the method, this paper uses (1) a method that does not use ASRMS to determine FDP points, (2) a method that does not involve temporal convolution, and (3) a multi-scale CNN method [31]. The method uses a short-time Fourier transform to pre-process the data, followed by a multi-scale feature extraction method in the middle of the network, and finally a fully connected layer for RUL prediction. The results are compared with those of the proposed method. All comparison methods are trained using the same training method, where the default parameters of the comparison methods (1), (2) are the same as the proposed method.

Figure 11 shows the degradation curves of the different methods on the same bearing. It is found that the proposed method is able to fit the degradation trend of the bearing much better than the other methods. However, the method without temporal convolution could not converge quickly at the end of the operation, and the error between the predicted values and label was large. Although the multi-CNN method can fit well at the end segment, the error is large at the beginning segment. The method without judging the FPD points has the disadvantages of both methods, which can neither fit the RUL labels in the beginning segment nor converge quickly at the end of the operation.

The results of the comparison with other methods are shown in Table 3. It can be observed that based on different evaluation metrics such as MAE, RMSE, the values of the proposed methods are lower than the methods used for comparison. This result shows that the proposed deep SG-TCN approach is able to extract the deep-level features of the high-dimensional data based on the correlation between multiple features, and thus can predict the degradation state of the bearings successfully.

## 5. Conclusions

In this paper, a novel spatial graph convolution-based model is proposed to predict the degradation process of bearings. Firstly, the first degradation point of the operating state is identified. Secondly, hybrid features are extracted from the original vibration signal, and then the extracted hybrid features are mapped to the spatial structure of the graph to compose the spatial graph structure of the bearing. Thirdly, the TCN is introduced to extract features for the temporal dimension of the spatial nodes of the bearing. Finally, the prediction of RUL is performed. The data of the PRONOSTIA platform is used to validate the proposed method, and the results show that the method has good performance in prediction.

On top of that, we should also note that the network structure of deep SG-TCN requires more time for training. Therefore, better hardware conditions are needed. Some degeneration states are abruptly changed bearings, which tend to fail at the end. Consequently, the identification of FDPs using ASRMS often can only get a relatively small number of training samples, which is unfavorable for deep learning. Hence, the method of depth-based learning needs to be explored further in the future.

## Figures and Tables

**Figure 1 sensors-21-04217-f001:**
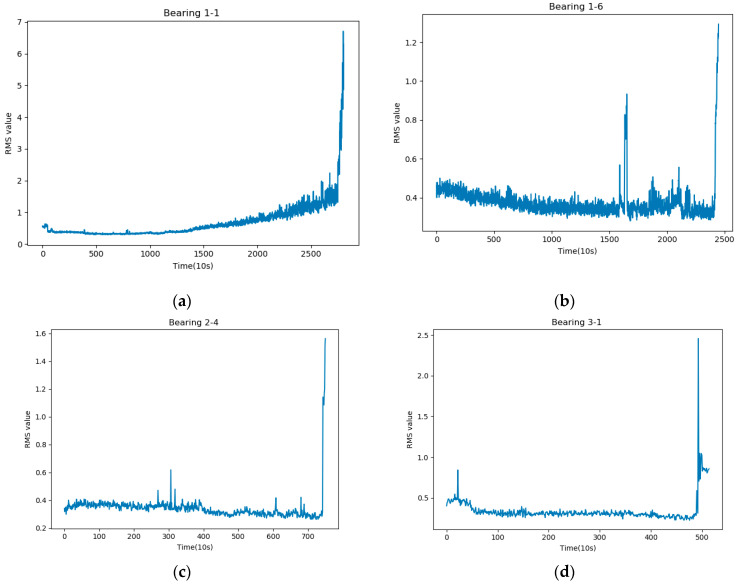
RMS values of bearings. Where (**a**–**d**) represents the RMS values for Bearing 1-1, Bearing 1-6, Bearing 2-4 and Bearing 3-1.

**Figure 2 sensors-21-04217-f002:**
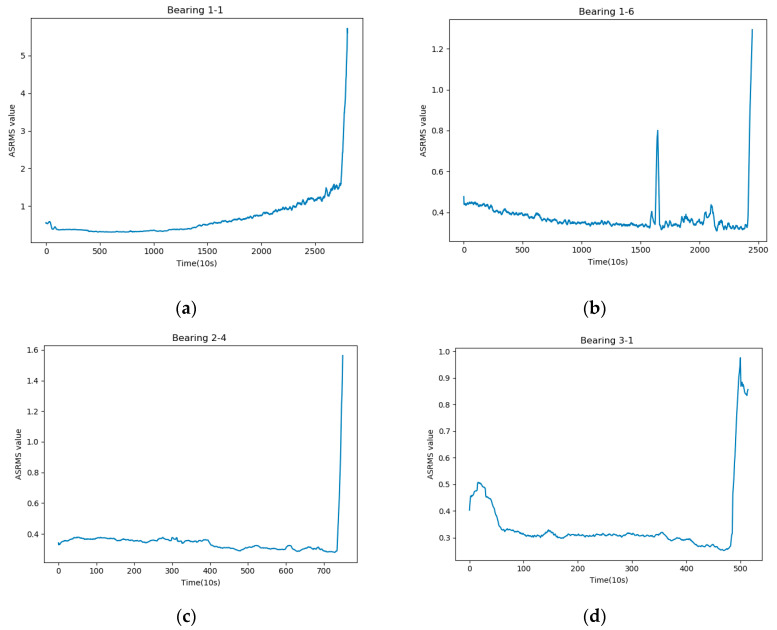
ASRMS curve of bearings. Where (**a**–**d**) represents the ASRMS curves for Bearing 1-1, Bearing 1-6, Bearing 2-4, and Bearing 3-1.

**Figure 3 sensors-21-04217-f003:**
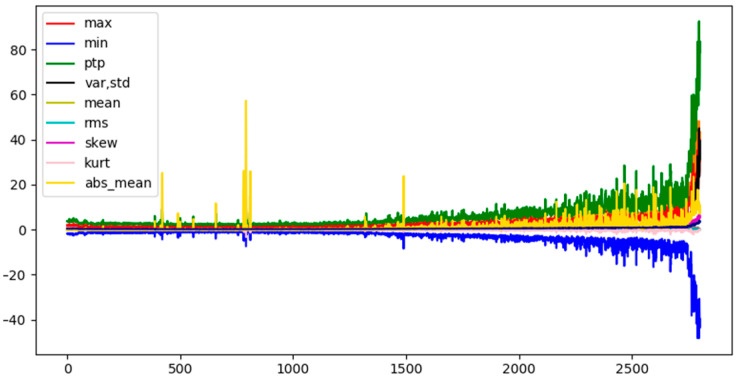
Tendency of selected 10 nodes feature.

**Figure 4 sensors-21-04217-f004:**
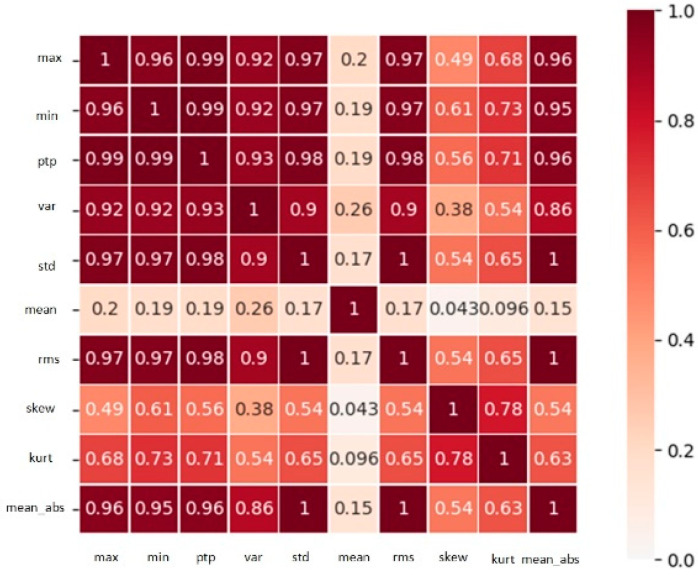
Bearing 1-3 correlation analysis heat map.

**Figure 5 sensors-21-04217-f005:**
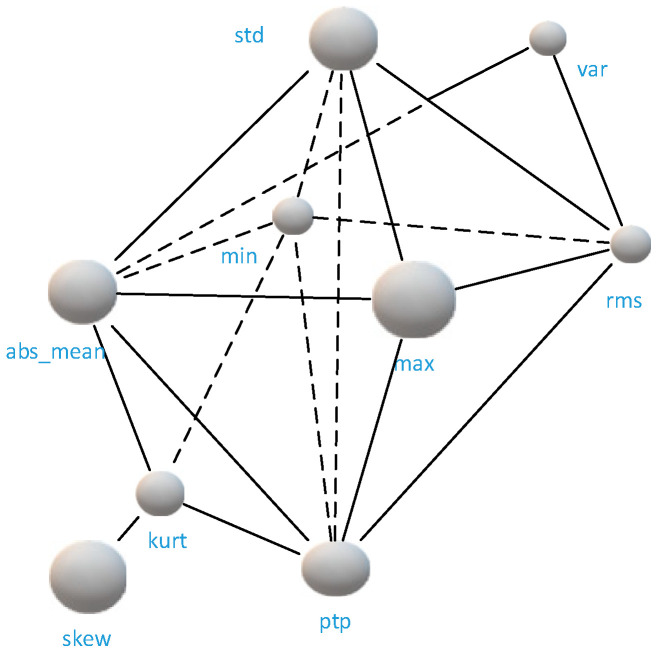
Spatial construction of the bearing.

**Figure 6 sensors-21-04217-f006:**
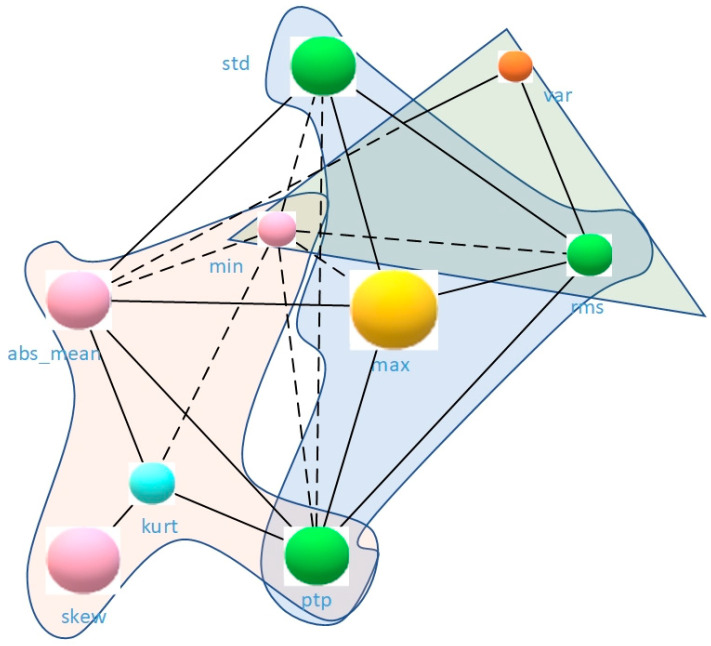
Spatial convolution of the bearings.

**Figure 7 sensors-21-04217-f007:**
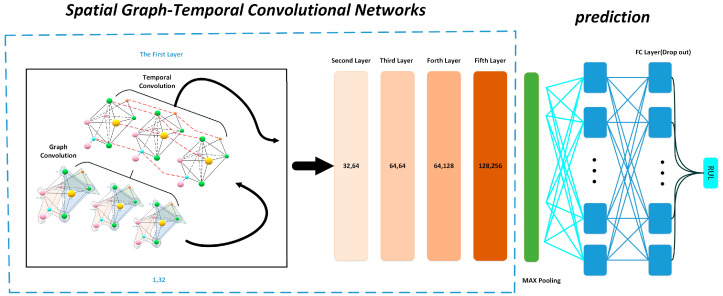
Network structure of SG-TCN.

**Figure 8 sensors-21-04217-f008:**
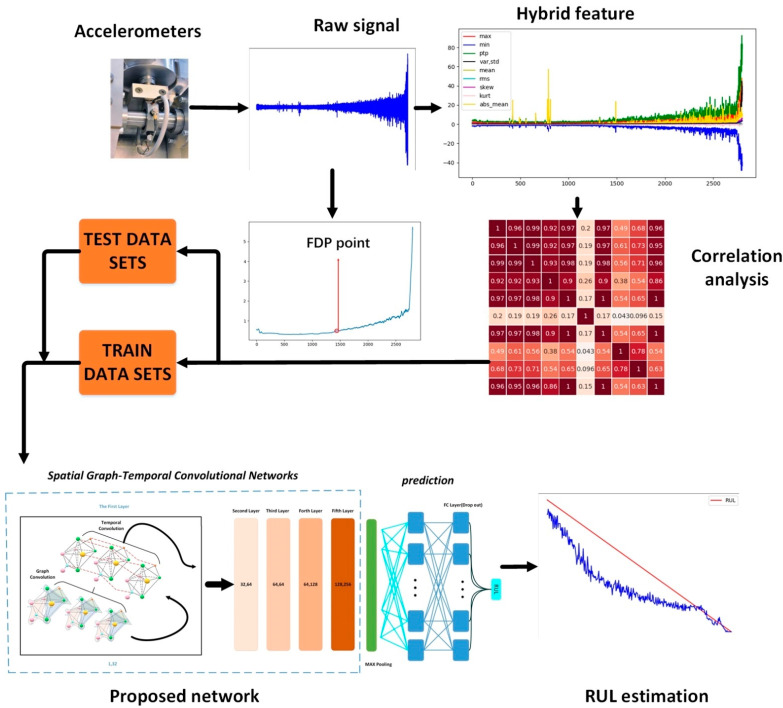
Flow chart of the proposed method.

**Figure 9 sensors-21-04217-f009:**
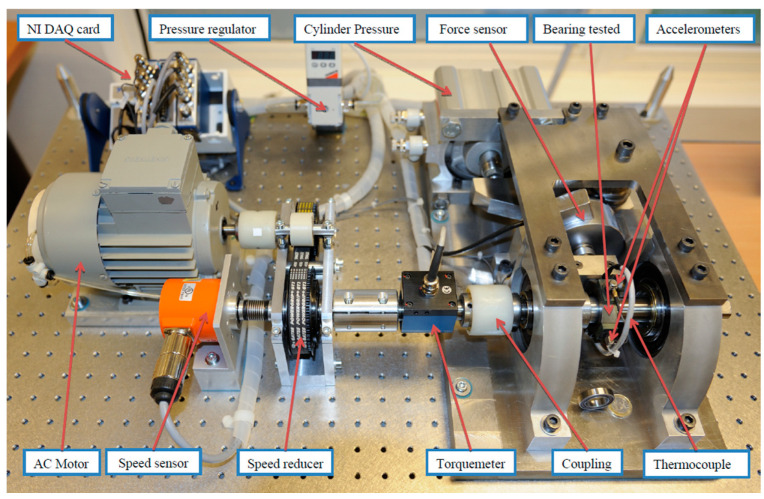
PRONOSTIA experimental platform.

**Figure 10 sensors-21-04217-f010:**
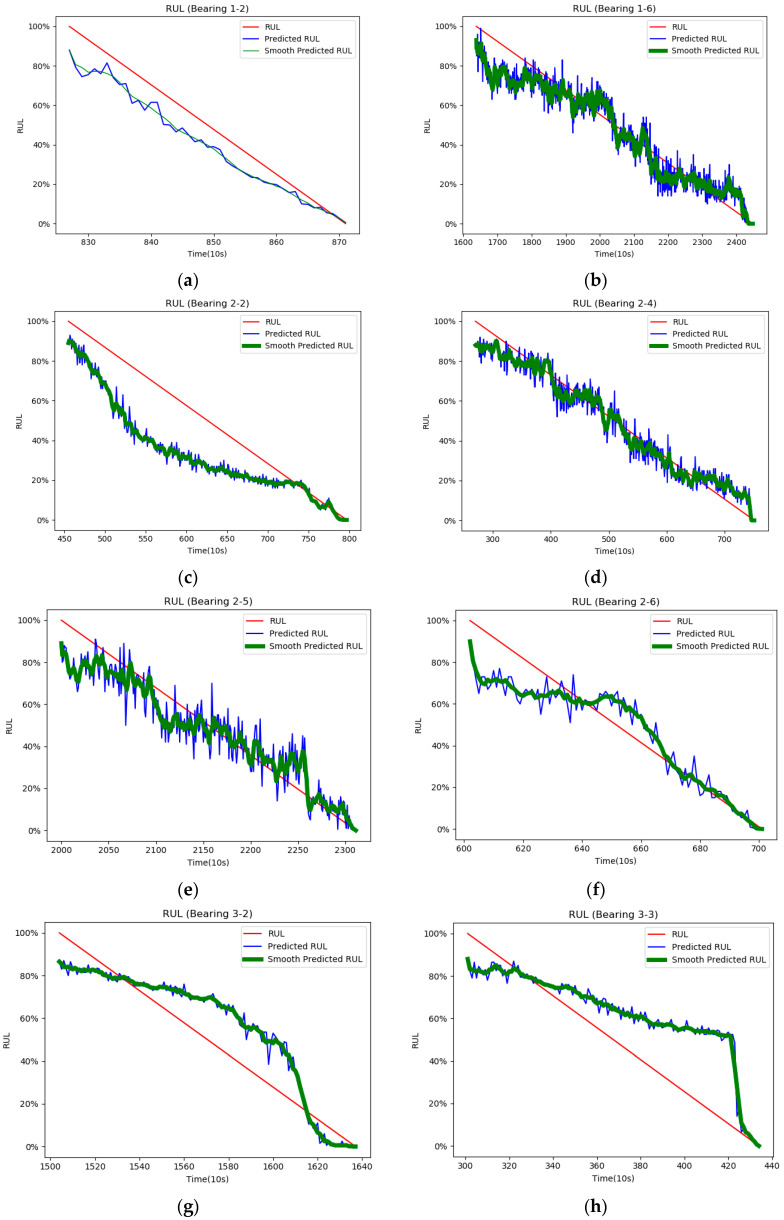
Test bearing RUL prediction results. Where (**a**–**h**) represents RUL prediction curves for each of the eight bearings.

**Figure 11 sensors-21-04217-f011:**
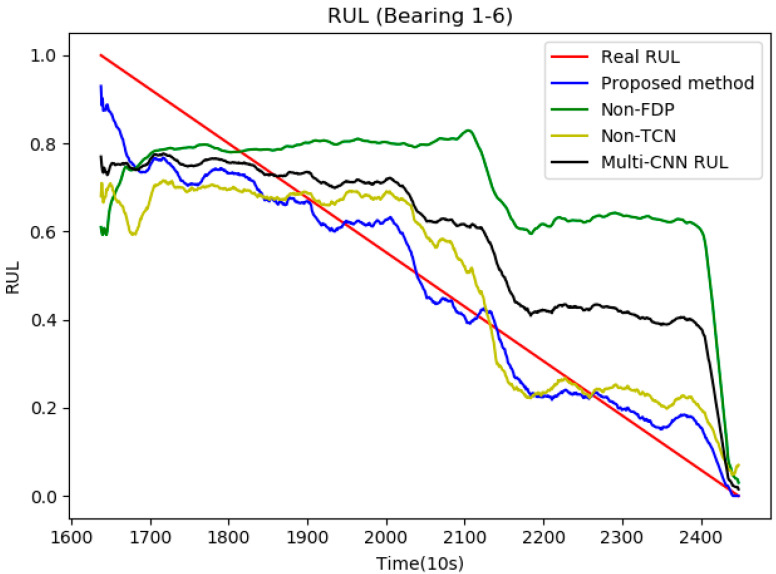
Prediction results of different methods for Bearings 1-6.

**Table 1 sensors-21-04217-t001:** Dataset information.

O.C. ^1^	Load(N)	Speed(rpm)	B.N. ^2^						
1	4000	1800	B1-1	B1-2	B1-3	B1-4	B1-5	B1-6	B1-7
2	4200	1650	B2-1	B2-2	B2-3	B2-4	B2-5	B2-6	B2-7
3	5000	1500	B3-1	B3-2	B3-3				

^1^ Operation Conditions. ^2^ Bearing Number.

**Table 2 sensors-21-04217-t002:** Prediction percentage error.

**Bearing 1-2**			**Bearing 1-6**			**Bearing 2-2**			**Bearing 2-4**		
**Act ^1^**	**Pre ^2^**	**Er ^3^**	**Act**	**Pre**	**Er**	**Act**	**Pre**	**Er**	**Act**	**Pre**	**Er**
450	396	12%	8110	7542	7%	3430	3052	11%	4820	4241	12%
419	353	13%	7519	6406	14.8%	3239	3018	6.8%	4489	4388	3.37%
388	366	5%	6928	6569	5.2%	3058	2641	13.7%	4308	4048	6%
357	319	10.7%	6588	6409	2.8%	2758	2160	21%	4108	3952	3.8%
337	281	16.7%	6017	5839	2.96%	782	651	16.7%	3908	3904	0.1%
306	276	9.8%	5436	5433	0.05%	762	686	9.9%	3727	3663	1.73%
276	209	24%	4926	4947	−0.4%	621	651	−1.5%	2455	2361	3.8%
245	186	23.9	4335	4460	−2.8%	581	583	−0.2%	2274	2265	0.41%
204	168	17.5%	3674	3649	0.68%	471	445	5.4%	1913	1879	1.8%
163	114	29.8%	3183	3162	0.66%	300	274	8.8%	1513	1494	1.25%
40	37	9.2%	2442	2433	0.41%	250	240	4.2%	1142	1108	3.0%
20	22	12.5%	310	243	21.6%	150	137	8.8%	521	530	−1.7%
**Bearing 2-5**			**Bearing 2-6**			**Bearing 3-2**			**Bearing 3-3**		
**Act**	**Pre**	**Er**	**Act**	**Pre**	**Er**	**Act**	**Pre**	**Er**	**Act**	**Pre**	**Er**
3120	2776	11%	1000	900	10%	1340	1159	13.5%	1340	1179	12%
2909	2620	9%	868	730	15%	1239	1112	10.3%	1229	1159	5.7%
2588	2464	3.27%	737	730	1%	1128	1092	3.22%	1138	1105	2.898%
2307	2308	−0.06%	636	570	10.4%	1047	1045	0.249%	1057	1065	−0.7%
2066	1716	16.1%	595	610	−4.12%	906	991	−9.36%	916	958	−4.5%
1755	1747	0.48%	444	500	−12.5%	795	958	−20.4%	795	897.8	−12.8%
1474	1497	−1.55%	333	360	−8.001%	685	904	−32%	695	824	−18.5%
1213	1216	−0.24%	292	280	4.414%	231	268	−15.6%	634	797	−25.6%
1033	967	6.3%	242	240	0.998%	221	201	9.3%	574	770	−34.2%
632	748	−18.5%	161	150	7.19%	201	167	16.9%	80	87	−8.1%
421	411	2.25%	101	85	15.85%	171	147	13.9%	50	67	−33%
30	15	48.2%	60	60	0%	60	34	44.58%	10	1	90%

^1^ actual RUL. ^2^ predicted RUL. ^3^ percentage error.

**Table 3 sensors-21-04217-t003:** Prediction preference comparisons with other methods.

Bearing Names.	SG-TCN		Non-FDP		Non-TCN		Multi-CNN	
	MAE	RMSE	MAE	RMSE	MAE	RMSE	MAE	RMSE
Bearing 1-1	16.4	21.7	51.3	60.45	21.9	26.4	31.7	35.7
Bearing 1-2	9.3	11.2	35.4	38.9	11.6	16.0	25.3	28.6
Bearing 1-3	6.1	8.8	23.8	31.7	15.5	18.7	20.6	24.5
Bearing 1-4	17.3	19.6	48.6	55.4	25.5	30.5	50.9	60.1
Bearing 1-5	14.1	17.8	43.5	50.8	18.9	24.6	29.4	34.6
Bearing 1-6	7.3	9.3	38.6	45.4	11.1	14.1	26.1	30.6
Bearing 1-7	6.6	8.9	20.4	24.7	10.6	14.2	16.2	19.3
Bearing 2-1	15.4	17.5	48.6	55.4	23.7	26.9	31.6	39.5
Bearing 2-2	16.4	19.6	64.5	90.5	20.5	25.4	32.2	50.8
Bearing 2-3	22.4	26.4	64.8	70.2	28.7	32.4	44.3	49.4
Bearing 2-4	5.6	6.8	30.5	33.4	10.1	13.4	18.4	21.5
Bearing 2-5	8.3	10.5	31.9	36.7	13.7	16.1	15.2	19.2
Bearing 2-6	15.5	18.3	46.7	49.3	21.4	24.1	31.6	36.9
Bearing 2-7	19.6	25.3	54.8	58.9	23.7	27.3	34.0	40.7
Bearing 3-1	15.4	19.6	49.3	56.1	26.1	29.4	32.4	36.2
Bearing 3-2	19.4	21.7	60.5	68.4	22.1	29.4	37.9	40.6
Bearing 3-3	23.3	28.6	70.1	75.8	27.9	43.6	45.8	78.4

## Data Availability

Restrictions apply to the availability of the dataset. PHM challenge datasets were provided by FEMTO-ST Institute (Besancon–France, http://www.femto-st.fr/). With the permission of the IEEE Reliability Society and FEMTO-ST Institute.

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
