# Peer review of "Remaining Useful Life Prognostics of Bearings Based on a Novel Spatial Graph-Temporal Convolution Network"

_sensors, 2021, doi:10.3390/s21124217_

Round 1

Reviewer 1 Report

I have read manuscript with great attention and interest. The article deals with an interesting topic and proposes a promising approach. Overall, the paper is well written and structured. However, in the text there are several grammatical  and typing errors (e.g., line 23: "as" should be "is"), so that the paper must be carefully proofreaded.

Author Response

Dear Editor,

Thank you very much for your valuable comments on the revision of this thesis. Based on your comments, I have corrected the errors you pointed out and made additional grammatical and formatting checks and corrections, all of which are included in the MS WORLD file. Please see the attachment.

Thank you again for your contribution to this paper!

Best regards

Reviewer 2 Report

This paper proposed a novel remaining useful life prognostics method based on average sliding root mean square, spatial graph convolution, and temporal convolution. The authors introduce this method in detail and applied it to a public dataset that is related to bearings. This paper is organized in a well-logical manner. However, it still has room for improvement. Some comments are listed below.

  1. You should revise the manuscript carefully due to some writing and formatting issues, for example:
  • In abstract, there is a formatting problem with the semicolon in Line 16.
  • In introduction, some citation marks, such as [24], [25], are added in wrong position. Please doublecheck and modify.
  • In section 2.1, the section name is “Average SRMS value”. Here the full name should be given when the abbreviation “SRMS” first appears, or you can directly rewrite it as “ASRMS value”.
  • In Line 240, the sentence “the convolution kernels are used using the residual block is concatenated to ensure” is not fluent.
  • In Line 282, the “( )” in formula should be changed to the “| |”.
  • In Line 332, the “It” in “In Table 2, It is observed that each bearing has…” does not need to be capitalized.
  1. This paper mentions “temporal convolution” and “time domain convolution” many times, please confirm if the above expressions have the same meaning. If so, please use a unified expression to avoid misunderstanding. If not, please explain the “temporal convolution network”proposed in Line 235.
  2. In Section 2.1, please clarify the basis for choosing 0.71 as the criterion of health stages.
  3. In section 4.3, the experiment results are comparied with some other methods to verify the effectiveness of the method. However, "multiscale CNN" is not mentioned above, and it is necessary to give a brief description or add some references.

Author Response

(The authors gave the same response as above.)
